# Synthesis of a Novel Semi-Conductive Composites Doping with La_0.8_Sr_0.2_MnO_3_ for Excellent Electric Performance for HVDC Cable

**DOI:** 10.3390/polym12040809

**Published:** 2020-04-04

**Authors:** Hongxia Yin, Yingcao Cui, Yanhui Wei, Chuncheng Hao, Qingquan Lei

**Affiliations:** 1Advanced Institute of Electrical Materials, Qingdao University of Science and Technology, Qingdao 266042, China; 2State Key Laboratory of Advanced Power Transmission Technology Global Energy Interconnection Research Institute co., Ltd., Beijing 102209, China

**Keywords:** HVDC cable, space charge accumulation, positive temperature coefficient effect, semi-conductive composites, La_0.8_Sr_0.2_MnO_3_

## Abstract

The semi-conductive layer located between the wire core and the insulation layer in high voltage direct current (HVDC) cable plays a vital role in uniform electric field and affecting space charges behaviors. In this work, the research idea of adding ionic conductive particles to semi-conductive materials to improve the conductive network and reduce the energy of the moving charge inside it and to suppress charge injection was proposed. Semi-conductive composites doped with different La_0.8_Sr_0.2_MnO_3_ (LSM) contents were prepared. Resistivity at different temperatures was measured to investigate the positive temperature coefficient (PTC) effect. Pulse electro-acoustic (PEA) method and thermal-stimulation depolarization currents (TSDC) tests of the insulation layers were carried out. From the results, space charge distribution and TSDC currents in the insulation samples were analyzed to evaluate the inhibitory effect on space charge injection. When LSM content is 6 wt. %, the experimental results show that the PTC effect of the specimen and charge injection are both being suppressed significantly. The maximum resistivity of it is decreased by 53.3% and the insulation sample has the smallest charge amount, 1.85 × 10^−7^ C under 10 kV/mm—decreased by 40%, 3.6 × 10^−7^ C under 20 kV/mm—decreased by 45%, and 6.42 × 10^−7^ C under 30 kV/mm—decreased by 26%. When the LSM content reaches 10 wt. %, the suppression effect on the PTC effect and the charge injection are both weakened, owing to the agglomeration of the conductive particles inside the composites which leads to the interface electric field distortion and results in charge injection enhancement.

## 1. Introduction

High voltage direct current (HVDC) cable exhibits remarkable properties in long-distance and large-capacity power transmission systems. However, there is a troublesome and urgent problem for the polyethylene insulation layer at present, namely space charge accumulation [1]. Space charge accumulation in the insulation layer can cause various problems such as partial discharges and premature insulation failure [2,3,4]. It is one of the key factors threatening the safe operation and restricting the development of HVDC cable. How to suppress the space charge injection from the conductive wire core to the insulation layer has caused wide concern [5,6,7,8].

Insulation materials such as low-density polyethylene (LDPE) and cross-linked polyethylene (XLPE) have been the focus of the research on space charge accumulation in the HVDC cable in the past twenty years [9,10,11,12]. Actually, the semi-conductive shielding layer lying between the conductive wire core and the insulation layer plays an important role in producing uniform high electric fields and preventing charge injection to the insulation layer, as the inevitable path for the migrating charges from the conductive wire core to the insulation layer. Usually, the traditional semi-conductive materials are manufactured with carbon black (CB) as the conductive particles, low-density polyethylene (LDPE) and ethylene-vinyl acetate copolymer (EVA) as polymer matrix [13,14,15,16,17,18]. In the actual application, the traditional semi-conductive composites often exhibit the positive temperature coefficient effect, as the resistivity of the semi-conductive composites increase as the temperature rises, increasing sharply near the melting region of the polymer, which is the positive temperature coefficient (PTC) effect [19,20]. The PTC effect will cause local heating of the semi-conductive shielding layer. As we know according to previous reports, the conductivity of the semi-conductive materials comes from the conductive network formed by the CB conductive particles, which are dispersed at the interface between the crystalline phase and the amorphous phase of the polymer matrix, where the coefficient of the thermal expansion of the polymer is much greater than that of the CB. When the temperature is rising, the polymer matrix expands and the distance between CB particles in the crystallization zone will increase with the change, correspondingly, so the conductive network will be broken down gradually. Thus, resistivity will increase gradually with the failure of the conductive network [21,22].

In recent years, there have some studies about the semi-conductive layers. The representative work was the study of the semi-conductive materials modified by the magnetic particle SrFe_12_O_19_. The results showed that space charge injection can be suppressed effectively as a small amount of SrFe_12_O_19_ was doped into the composites while the PTC effect cannot be suppressed effectively at the same SrFe_12_O_19_ content [23].

The perovskite compound LaMnO_3_ is a cubic crystal system with a face-centered cubic lattice. After part of La^3+^ ions are replaced by Sr^2+^ ions in LaMnO_3_, to maintain electric neutrality, part of Mn^3+^ ions are converted to Mn^4+^ ions. During this conversion, a small polaron Mn^3+^–O^2−^–Mn^4+^ is produced. In contrast from the CB carriers, its concentration is only related to the concentration of doped ions, and has nothing to do with temperature. However, the migration of the polaron changes with temperature change, i.e., it can be excited by heating, to make the d-electron vacancy jump from the Mn^4+^ ion to the Mn^3+^ ion and correspondingly electrons (emitted from the d-band) jump from Mn^3+^ ions to Mn^4+^ ions. Through the migration of the small polarons charge transfer becomes easier, which brings superior conductivity for La_0.8_S_r0.2_MnO_3_. The structure of La_0.8_S_r0.2_MnO_3_ is shown in Figure 1. As a super ionic conductor, La_0.8_Sr_0.2_MnO_3_ has got a lot of attention in the field of solid oxide fuel cells (SOFC) due to its excellent ionic conductivity [24,25]. 

In this work, the research idea of adding ionic conductor La_0.8_S_r0.2_MnO_3_ to the semi-conductive composites to improve the conductive network inside them, and to reduce the energy of the moving charges in the semi-conductive layer to suppress charge injection was proposed, as shown in Figure 2.

## 2. Materials and Methods

### 2.1. Materials

La(NO_3_)_3_·6H_2_O was obtained from Damao Chemical Co. Ltd. (Tianjin, China); Sr(NO_3_)_2_, Mn(NO_3_)_3_ (50% aqueous solution) and ammonia water were purchased from Sinopharm Chemical Co. Ltd. (Shanghai, China); citric acid was obtained from Guangfu Technology Development Co. Ltd. (Tianjin, China); these reagents were of analytical reagent grade. LDPE (Kunlun 18D) was obtained from the National Petroleum Corporation (Daqing, China); ethylene-vinyl acetate copolymer (EVA,7470) was obtained from Formosa Plastics Co. LTD (Taiwan, China) and conductive carbon black (CB, VXC-72) was purchased from CABOT Co. LTD (Boston, MA, USA)

### 2.2. Preparation of La_0.8_Sr_0.2_MnO_3_ Nano-Powder

#### 2.2.1. Synthesis of La_0.8_Sr_0.2_MnO_3_

La_0.8_Sr_0.2_MnO_3_ was synthesized by sol-gel-high temperature sintering method. The precursor materials La(NO_3_)_3_·6H_2_O: Sr(NO_3_)_2_: Mn(NO_3_)_3_ in a stoichiometric ratio of 0.8:0.2:1 were dissolved in deionized water and 1 mol·L^−1^ mixed solution was prepared. Citric acid equivalent to 1.2 times of all the metal ions was dissolved in deionized water too. The citric acid solution was then added into the mixed solution to form a metal complex compound. Ammonia water was added dropwise to adjust the pH of the solution to 6. The solution was heated continuously in an oil bath with stirring at 80 °C until most of the water was removed and the sol turns into gel. Then the gel was dried in a dry oven at 200 °C until it became a white solid. Then the solid was put into a Muffle furnace (SGM, M8/14A, Sigma Instrument Manufacture Co. Ltd., Shanghai Branch, Shanghai, China.) carbonized for 1 hour at 400 °C at first, then temperature was raised to 1000 °C and sample was sintered for 4 h [26].

#### 2.2.2. Ball Milling of La_0.8_Sr_0.2_MnO_3_

Next, the synthesized perovskite material La_0.8_Sr_0.2_MnO_3_ was ball milled in a planetary ball miller (MiQi YXQM-4L, Changsha Miqi Equipment Co. Ltd., Changsha, China.) at various time durations, and the effect of the different durations of ball milling were investigated by the X-ray diffraction (XRD, D-MAX 2500/PC, Rigaku, Tokyo, Japan.) and field emission scanning electron microscopy (FE-SEM, JSM-6700F, JEOL, Tokyo, Japan). According to Cheng X’s previous work, for the ball milling process, the sample was put into the ball milling reactors and zirconia balls with different sizes were added in the powder in the weight ratio of 1:20. For wet milling, anhydrous ethanol was added into the powder up to three quarters of the volume of the reactor. The ball milling process was performed at 300 rpm for 12, 24, 36, 48 and 60 h respectively, and samples were dried at 60 °C for 24 h after being ball milled [27]. 

### 2.3. Synthesis of La_0.8_Sr_0.2_MnO_3_/Semi-Conductive Composites

The La_0.8_Sr_0.2_MnO_3_/semi-conductive composites were prepared by the melt-compounding method in three steps. Firstly, LDPE, EVA and CB were pre-mixed together in a certain proportion at 150 °C with a two-roll miller (HaPro Electric Co. Ltd., Harbin, China). Secondly, LSM nano-powder was mixed together with the pre-mixed material in a mixed torque rheometer (RM-200C, HaPro Electric Co. Ltd., Harbin, China) for 12 min in a ratio of 0, 2, 4, 6, 8 and 10 wt. % at 120 °C. Thirdly, the mixed materials with different contents of La_0.8_Sr_0.2_MnO_3_ were placed in the mold and pressed into a sheet by the plate vulcanizing press (SKZ401, Jinan Qingqiang Electromechanical Equipment Co., Ltd., Jinan, China) at 130 °C for 6 min with the pressure of 10 MPa. Then the cross-linked process was carried out by the plate vulcanizing press at 180 °C for 15 min. Lastly, the specimen was cooled for 10 min with 10 MPa.

### 2.4. Electric Performance Tests

Prior to the tests, the specimens were cleaned with anhydrous ethanol and placed in dry oven at 60 °C for 6 h.

#### 2.4.1. Resistivity-Temperature Test

Resistivity of La_0.8_Sr_0.2_MnO_3_/semi-conductive composites was measured at different temperatures by a semi-conductive resistance test device. Considering the working condition of the HVDC cable, in the experiment, 10 typical temperatures of 30, 40, 50, 60, 70, 80, 90, 100, 110 and 120 °C were set.

#### 2.4.2. Pulse Electro-Acoustic (PEA) Method

Space charges in the HVDC cable’s insulation layer were mainly composed of homo-charges generated by the injection of the electrode and the hetero-charges generated by the ionization of impurities [28]. When the electrical field reaches about 10 kV/mm, the injected charges from the electrodes are dominant, which are the main source of the space charges in the insulation layer.

In order to observe the influence of La_0.8_Sr_0.2_MnO_3_/semi-conductive composites to space charge distribution in the insulation material, the pulse electro-acoustic (PEA) method was used. For the method, the La_0.8_Sr_0.2_MnO_3_/semi-conductive layer between the metal electrode and the insulation material plays the role of acoustic impedance matching, meanwhile it would affect the charge injection in the insulation material [29]. A high voltage electric pulse was applied through the semi-conductive to the insulation layer to cause a small movement of the space charges in the insulation medium. This tiny displacement was propagated to the piezoelectric sensor as a sound wave and was collected and analyzed. Therefore, space charge distribution in the insulation material can be observed intuitively. The schematic of the PEA method is shown in Figure 3.

In the experiments, semi-conductive composites with different La_0.8_Sr_0.2_MnO_3_ concentrations were used as the test sample. The applied electric fields were 10, 20 and 30 kV/mm, keeping time for 1800 s.

#### 2.4.3. Thermally Stimulated Depolarization Current Test 

Thermally stimulated theory has been a mature method to study the trap and charge characteristics of polymers which can also reflect charge accumulation and distribution in the material [30,31]. In order to verify the influence of La_0.8_S_r0.2_MnO_3_ content on charge injection to the insulation medium under strong electric field, the thermally stimulated depolarization current of the insulation sample was measured. The test involved two steps. At first, the polarization process was carried using a self-built system containing a pair of cylindrical copper electrodes with the diameter of 25 mm, a high voltage DC source (Dongwen High Voltage Power Supply Tianjin Co., Ltd., Tianjin, China) and a shielding box outside as is shown in Figure 4a. The La_0.8_S_r0.2_MnO_3_/semi-conductive layer was placed between the upper electrode and the insulation layer as the shielding layer. The thickness of LSM/semi-conductive sheet was 0.5 mm. The insulation sample was a LDPE round sheet with a uniform internal structure, thickness of 0.3 mm, and its surface was cleaned with anhydrous ethanol and placed in a drying oven at 60 °C for 4 h. In the experiment, three typical electric fields, 10, 20 and 30 kV/mm were chosen, polarizing for 30 min. 

After removing the voltage, the depolarization current process was carried out by the Novocontrol system (Novocontrol Technologies GmbH & Co. KG, Frankfurt, Hwsse-Damstadt, Germany.) to measure the depolarization currents of the insulation samples. The current was measured in a small size vacuum chamber, and the temperature can be controlled in adjustable rate. The schematic of the Novocontrol system is shown in Figure 4b. In this test, the heating range changed from 20 to 90 °C, and the heating rate was set to 5 °C/min.

### 2.5. Structural Characterization of Ball Milled La_0.8_Sr_0.2_MnO_3_ and La_0.8_Sr_0.2_MnO_3_/Semi-Conductive Composites

Structural analysis of the La_0.8_Sr_0.2_MnO_3_ ball milled for different durations was carried out with X-ray diffract meter (D-MAX 2500/PC, Rigaku, Tokyo, Japan) at 18 kW, 40 kV, and 40 mA using Cu Kα radiation (1.54059 Å) at room temperature with the scanning range of 10~80°. The micromorphology of La_0.8_Sr_0.2_MnO_3_nano powder was observed by the field emission scanning electron microscopy JSM-6700F, JEOL, Tokyo, Japan). 

The specimens with La_0.8_Sr_0.2_MnO_3_ content of 2, 6 and 10 wt. % were frozen in liquid nitrogen, and sliced. The cross-section micromorphology of them were observed by SEM (FEI/Nova Nano SEM 450, FEI Company, Hillsboro, OR, USA). While the SEM was used, an elemental energy spectrum analysis can be performed, and an elemental Energy Dispersive Spectrum (EDS, FEI/Nova Nano SEM 450, FEI Company, Hillsboro, OR, USA) analysis can be performed on the same specimen. The cross-section micromorphology analysis is a common method for rubber-plastic composites.

## 3. Results

### 3.1. Structure Analysis of La_0.6_Sr_0.4_MnO_3_ Nano-Powder

The XRD patterns of La_0.8_Sr_0.2_MnO_3_ powder with different ball milling durations (0, 12, 24, 36 48 and 60 h) are shown in Figure 5. 

As is shown in Figure 5, samples of different ball-milling duration show characteristic peaks of La_0.8_Sr_0.2_MnO_3_ at 2θ = 22.8°, 32.3°, 39.9°, 40.4°, 46.6°, 58.4°, 67.7°, 68.4°, 77.1 and 77.7°, which correspond to the hkl planes of (110), (020), (022), (202), (220), (312), (040), (400), (240) and (332) respectively. All curves are in accordance with JCPDS No. 53-0058. There are no impurity peaks detected in the patterns which indicates that the synthesizing method of La_0.8_Sr_0.2_MnO_3_ is proper. According to Figure 5, the diffraction patterns of La_0.8_Sr_0.2_MnO_3_ powder with different ball milling durations show difference only in the peak intensity, which indicates that the crystal structure of La_0.8_Sr_0.2_MnO_3_ has not been damaged by ball milling.

The FE-SEM images for the morphology of La_0.8_Sr_0.2_MnO_3_ nano powder with different ball-milling durations are shown in Figure 6. The images in Figure 6a–f are corresponding to ball milling times of 0, 12, 24, 36, 48, and 60 h, respectively. 

As is shown in Figure 6, La_0.8_Sr_0.2_MnO_3_ without being ball milled has large particles and uneven sizes. When it was ball milled for 36 h, there were only a few large crystals. Until being ball milled for 48 h, the particles become uniformly sized and large particles disappeared, with most of the particles are between 100 and 200 nm. After being ball milled for 60 h, grain size was almost the same as that of ball milling for 48 h which indicates that the proper ball milling time for La_0.8_Sr_0.2_MnO_3_ is 48 h.

It can be concluded from Figure 5 and Figure 6 that the sintered La_0.8_Sr_0.2_MnO_3_ sample can turned into a nano-powder with sizes of 100–200 nm by ball milling at 300 rpm for 48 h.

### 3.2. Cross-Section Micromorphology of the LSM/Semi-Conductive Composites

The specimens were frozen and sliced in liquid nitrogen, and the microstructure of the cross section was observed by SEM and EDS analysis was performed on the same specimen.

Figure 7a–c shows SEM images of the cross sections of La_0.8_Sr_0.2_MnO_3_/semi-conductive composites with La_0.8_Sr_0.2_MnO_3_ contents of 2, 6 and 10 wt. %, respectively. It can be observed from Figure 7a–c that La_0.8_S_r0.2_MnO_3_ particles are well-dispersed in the semi-conductive matrix when content is 2 and 6 wt. %. When the La_0.8_Sr_0.2_MnO_3_ content is 10 wt. %, particles become larger, which indicates agglomeration of La_0.8_Sr_0.2_MnO_3_ occurs. Figure 7d,e shows the element analysis of the specimens. The test area has been circled by red oval as is shown in Figure 7d. It can be seen from Figure 7e that the content of C is the highest of the test area, followed by the content of O, the contents of Mn, and La and Sr is the next. The test result is consistent with the SEM observation of the test area in Figure 7d. It can be confirmed that La_0.8_Sr_0.2_MnO_3_ particles were well blended with the semi-conductive matrix. 

### 3.3. Resistivity of La_0.8_Sr_0.2_MnO_3_/Semi-Conductive Composites

Figure 8 shows the temperature-resistivity behaviors for the semi-conductive composites. It can be seen from Figure 8a that the original resistances of the specimens are basically consistent. Upon raising the temperature, all the specimens show an ascending trend in resistance and a pronounced PTC effect occurs around the melting point of the polymer matrix. After reaching the peak, resistivity begins to decrease as the temperature continues to rise. The specimen shows the negative temperature coefficient (NTC) effect owing to the agglomeration of conductive particles caused by changes in the microstructure of the polymer matrix. 

The melt range of LDPE is 110–115 °C and the melt point of EVA 7470 is 76 °C. Furthermore, during the preparation of the semi-conductive composite, we also learned that the polymers have melted at 120 °C. Moreover, considering its insulation is polyethylene, the operating temperature of the HVDC cable cannot exceed 363 K [5,6] since the NTC effect occurs at 393 K which is much higher than the operational temperature of the HVDC cable. Thus, the phenomenon can be ignored for HVDC cable. 

It should be noted that the PTC effect of La_0.8_Sr_0.2_MnO_3_/semi-conductive composites is significantly decreased compared to that without La_0.8_Sr_0.2_MnO_3_. Figure 8b depicts the correlation of the maximum resistance as a function of the La_0.8_Sr_0.2_MnO_3_ concentration. It can be seen that the maximum resistivity of the semi-conductive composites with 6 wt. % La_0.8_Sr_0.2_MnO_3_ is the smallest, only 202 Ω cm, which decreased by 53.3% compared to the specimen without La_0.8_Sr_0.2_MnO_3_. It also can be seen from Figure 8a that the PTC effect for the specimen without La_0.8_Sr_0.2_MnO_3_ occurs at 363 K, whereas, it occurs at 373 K for the La_0.8_Sr_0.2_MnO_3_/semi-conductive composites. We have known that the operating temperature of the HVDC cable cannot exceed 363 K, which means no obvious PTC effect will occur in the operating temperature for the HVDC cable, and that the La_0.8_Sr_0.2_MnO_3_/semi-conductive will not have obvious PTC effect, if it is of great significance for safe HVDC cable operation. 

### 3.4. Space Charge Distribution

Space charge distribution in the insulation layer at 1800 s under the action of the semi-conductive layer with different La_0.8_S_r0.2_MnO_3_ content are compared, as is shown in Figure 9.

When keeping the same applied electric field, the induced charges near the electrode will not change. Therefore, the difference of the charges near the electrode mainly come from the injected charge. It can be seen from Figure 9a–c that when shielded by the semi-conductive layer with La_0.8_S_r0.2_MnO_3_ content of 6 wt. %, the charge near the electrodes is minimum and the charge amount decreased obviously when compared to the specimen without La_0.8_S_r0.2_MnO_3_, and then increases with the increase of La_0.8_S_r0.2_MnO_3_ content. 

It is shown in Figure 9d that the accumulated charges in the insulation layer firstly decreased and then increased with the increase of La_0.8_S_r0.2_MnO_3_ concentration. When La_0.8_S_r0.2_MnO_3_ content is 6 wt. % the charge amount is the smallest, and it is 1.85 × 10^−7^ C under 10 kV/mm, 3.6 × 10^−7^ C under 20 kV/mm, and 6.42 × 10^−7^ C under 30 kV/mm, decreased by 40%, 45%, and 26% respectively. It indicates that semi-conductive composites with La_0.8_S_r0.2_MnO_3_ content 6 wt. % can inhibit the charges injecting in the insulation layer more effectively, especially under the stress of 20 kV/mm.

When the La_0.8_S_r0.2_MnO_3_ content exceeds 6 wt. %, the charge suppression is weakened, and the charge amount of the insulation layer begins to increase. Because most of the inorganic conductive particles can only be distributed at the interface between the crystalline phase and the amorphous phase, thus, too many inorganic particles inside the polymer matrix will agglomerate, accordingly. Oversized particles can cause surface roughness inside the semi-conductive, which will lead to the interface electric field distortion and result in charge injection enhancement.

### 3.5. Thermally Stimulated Depolarization Current (TSDC) Tests

Figure 10a–c shows the depolarization currents in the insulation samples under the polarization stress of 10, 20 and 30 kV/mm respectively. It can be seen that the depolarization currents in the insulation samples are significantly different due to the La_0.8_Sr_0.2_MnO_3_ content. When La_0.8_Sr_0.2_MnO_3_ content is 6 wt. %, the peak current of the insulation sample is the smallest, 4.5 × 10^−12^ A under polarization stress of 10 kV/mm, decreased by 37%, 1.1 × 10^−12^ A under polarization stress of 20 kV/mm, decreased by 21%, and 6.2 × 10^−12^ A under polarization stress of 30 kV/mm, decreased by 37%.

In addition, it can be seen from Figure 10 that the depolarization currents in the insulation sample polarized under 20 kV/mm are much lower than other polarized stresses. When polarization stress is 20 kV/mm, the small polarons of La_0.8_S_r0.2_MnO_3_ are polarized further compared to which under the stress of 10 kV/mm. This accordingly causes the electric cloud of the small polarons being distorted more, thus Coulomb force becomes stronger. This means the moving electrons will be blocked by a stronger Coulomb force in the semi-conductive layer and it becomes more difficult to hop through the interface between the semi-conductive layer and the insulation layer. Thus, charges injected into the insulation layer are suppressed more effectively. Since the deformation of the lattice is limited, there is a maximum force between the cations and the moving electrons. When polarize stress reaches 30 kV/mm, the action of the electric field to the electron has significantly exceeded the action of the Coulomb force, therefore, the peak value of the depolarization currents increases obviously. In summary, the La_0.8_Sr_0.2_MnO_3_/semi-conductive composites can suppress space charge injection to the insulation layer most effectively under electric field 20 kV/mm. It is notable that the working condition of the HVDC cable is just 20 kV/mm.

## 4. Discussion

It is indicated from the resistivity experimental results that the PTC effect of La_0.8_Sr_0.2_MnO_3_/semi-conductive composites has been suppressed more effectively, which means that La_0.8_Sr_0.2_MnO_3_ does form an improved conductive network inside the semi-conductive composites. According to the previous report, it can be obtained that inorganic particles and polymers could be regarded as having three-phase coexistence at low temperature, i.e., a crystalline phase, an amorphous phase, and an inorganic particle phase. Most of the inorganic conductive particles could only be dispersed at the interface between the crystalline phase and the amorphous phase in which the molecular chains are randomly arranged and the conductive particles could not enter the crystalline region of the polymer. At room temperature, the concentration of the conductive particles is higher compared to LDPE/EVA composites, therefore, particle to particle contact exists between the distributed conductive particles thus giving the total composition a relatively low resistance after the conductive frame has been formed. When the temperature is lower than the melting point of the polymer the movement of the macromolecular chain is not free, hence, the resistivity of the material raised slowly. When the temperature raises near to the melting point of the polymer, the polymer begins to melt, the crystal phase changes to amorphous phase, and the conductive particles gradually enter the amorphous phase region converted a moment ago. Thus, the conductive particles were diluted and the distance of the conductive particles increased. The conductive paths formed at low temperature begin to break, then the resistivity of the composites increases sharply near the melting point. This is the reason for the generation of the PTC effect. When the temperature continues to rise until it is higher than the melting point of the composite, the resistivity begins to decrease again and the NTC phenomenon occurs. Because of the fluidity of the system being enhanced after the polymer’s melting, the conductive particles obtain higher energy which enables them to move in a larger range. At the same time, the inorganic nanoparticles have stronger agglomeration force, which is more prone to dispersion-aggregation phase transition. The separated conductive particles agglomerate again and form a new conductive network which caused the NTC effect [20,21,22,32].

The schematic diagram of the forming of the conductive network in different semi-conductive composites is shown in Figure 11. The conductive network of the semi-conductive composites without La_0.8_S_r0.2_MnO_3_ formed only by CB is shown in Figure 11a. With the temperature rising, the polymer matrix begins to expand, and the distance between the conductive particles begins to increase. As the temperatureis increasing, more and more conductive particles cannot contact each other, therefore the conductive network is gradually being destroyed, as is shown in Figure 11b. When La_0.8_S_r0.2_MnO_3_ is doped in the composites and is finely dispersed, the small polarons will be excited with heating. According to the previous report, the polarons can not only contribute electron transitions, but also the corresponding number of hole transitions [25], thus, many more new conductive channels are formed, and accordingly the specimen can maintain a certain conductivity. When La_0.8_Sr_0.2_MnO_3_ content is 6 wt. % the new network is the most effective which means the small polarons of La_0.8_S_r0.2_MnO_3_ and electrons of carbon black can compensate each other to form the most effective conductive network as is shown in Figure 11c. Even if the temperature rises, due to the presence of small polarons, the fracture of the conductive network will be weakened. As is shown in Figure 8, the PTC effect of the La_0.8_S_r0.2_MnO_3_ doped specimen was reduced at various degree. In according with the previous study, the inorganic particles could only be scattered at the interface between the crystalline phase and the amorphous phase [20,22]. Thus, when there are too many inorganic La_0.8_Sr_0.2_MnO_3_ particles among the polymer matrix, for example, 10 wt. %, they are easier to agglomerate and oversized particles will be formed. It also can be observed from Figure 7c that the particles have agglomerated together and the particle size is more bigger than that in Figure 7a,b. Particle agglomeration will reduce the effective doping concentration of La_0.8_Sr_0.2_MnO_3_ and lead to an increase of large particles, which will cause interfacial effects, and also hinder the migration of the small polarons and electrons. As a result, the conductivity of the sample is reduced, while the temperature rising leads to part of the conductive channels being blocked again, as is shown in Figure 11d. As a result, the PTC effect becomes more obvious again.

When the semi-conductive composites is without La_0.8_Sr_0.2_MnO_3_ doping, under the strong electric field, the free charge in the metal electrode will hop through the interface barrier and migrate towards the insulation material. The moving charges are only affected by the electric field, and the trajectory of them is approximately straight [23], as is shown in Figure 12a.

We have known that La_0.8_Sr_0.2_MnO_3_ particles can only dispersed at the interface between the crystalline phase and the amorphous phase. When the La_0.8_Sr_0.2_MnO_3_ concentration is not high, for example no more than 6 wt. %, particles can be distributed evenly, just as is shown in Figure 7a,b. In accordance with the rules of outside nuclear electronic arrangement, the d-band of both Mn^3+^ and Mn^4+^ have not reached half full, they both have multiple empty d-bands, so they have a strong attraction for free electrons. When moving charges entered the semi-conductive layer, the charges will be affected not only by the electric field but also attracted by the Mn^3+^ and Mn^4+^ ions which is the Coulomb force [23,25], as is shown in Figure 12b.

The moving charges suffer resistance along the vertical direction, which is related to the concentration of the small polarons of La_0.8_Sr_0.2_MnO_3_. The experimental results show that when La_0.8_Sr_0.2_MnO_3_ content exceeds 6 wt. %, the effect of the La_0.8_Sr_0.2_MnO_3_/semi-conductive composites on inhibiting electron injection to the insulation layer is weakened and even leads to the increase of electron in the insulation sample. The peak of the depolarization current in the insulation sample for 10 wt. % La_0.8_Sr_0.2_MnO_3_ content also becomes high, because most of the inorganic conductive particles can only be distributed at the interface between the crystalline phase and the amorphous phase, and too many inorganic particles inside the polymer matrix will lead to agglomeration, as is shown in Figure 12c. Accordingly, the surface roughness between the conductive materials and polymers in the semi-conductive increases, which leads to the interface electric field distortion and results in charge injection enhancement. 

## 5. Conclusions

In this work, the research idea of adding the ionic conductor La_0.8_S_r0.2_MnO_3_ to the semi-conductive composites to build a new conductive network to suppress the PTC effect and to reduce the energy of the moving charges in the semi-conductive layer to suppress charge hop through the interface barrier and migrate towards the insulation material was proposed. The influence of the semi-conductive polymers modified with La_0.8_S_r0.2_MnO_3_ on the PTC effect suppress characteristics and charge injection characteristics in the insulation sample were studied. The experimental results show that a certain amount of La_0.8_S_r0.2_MnO_3_ can significantly suppress the PTC effect and charge injection. The conclusions are drawn as follows:

(1) When the La_0.8_S_r0.2_MnO_3_ content is 6 wt. %, the experimental results show that the PTC effect and charge injection are both being suppressed most effectively. The lowest maximum resistivity is 202 Ω cm, decreased by 53.3% compared to the specimen without La_0.8_S_r0.2_MnO_3_, and the insulation sample has the smallest charge amount, 1.85 × 10^−7^ C under 10 kV/mm—decreased by 40%, 3.6 × 10^−7^ C under 20 kV/mm—decreased by 45%, and 6.42 × 10^−7^ C under 30 kV/mm—decreased by 26%. In addition, the peak of the depolarization current is the minimum, 4.5 × 10^−12^ A under 10 kV/mm, 1.1 × 10^−12^ A under 20 kV/mm, 6.2 × 10^−12^ A under 30 kV/mm, decreased by 37%, 21% and 37% respectively. Because the moving charges are affected by the Coulomb force from the polarized small polaron, in addition, the vertical component of which will offset part of the electric field force and the horizontal component of which will deflect the charge migration path in LSM/semi-conductive composites. This will lead to the electrons moving towards the insulation sample being blocked and quite a lot of electrons being unable to enter the insulation sample.

(2) When the La_0.8_Sr_0.2_MnO_3_ content is 10 wt. %, the PTC effect suppression is not as obvious as that of 6 wt. % and the charge suppression effect is weakened. This is because too many inorganic particles inside the polymer matrix will agglomerate, which causes surface roughness of the semi-conductive composites, which causes the interface electric field distortion and charge injection to be enhanced again.

## Figures and Tables

**Figure 1 polymers-12-00809-f001:**
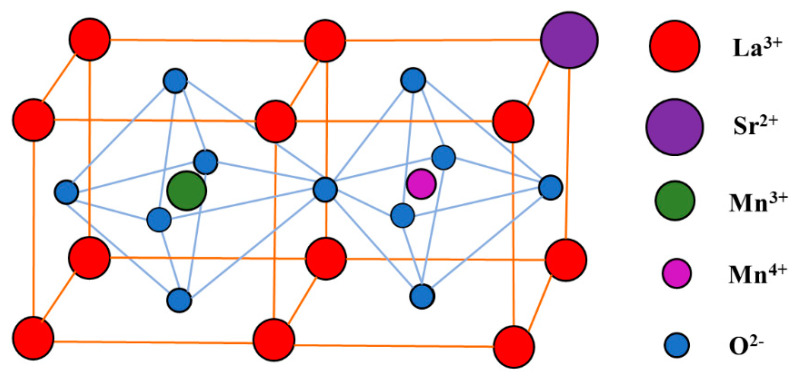
Structure of La_0.8_Sr_0.2_MnO_3_.

**Figure 2 polymers-12-00809-f002:**
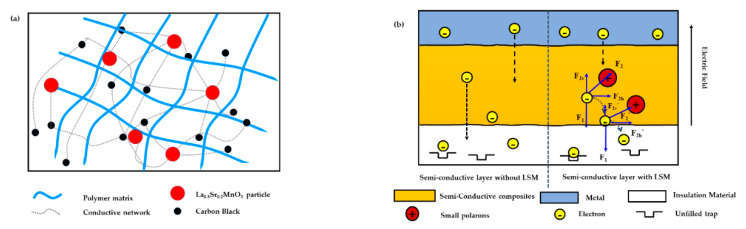
Schematic diagram of research idea. (**a**) Conductive network of the semi-conductive layer with La_0.8_Sr_0.2_MnO_3_ (LSM); (**b**) electrons moving path in the semi-conductive layer.

**Figure 3 polymers-12-00809-f003:**
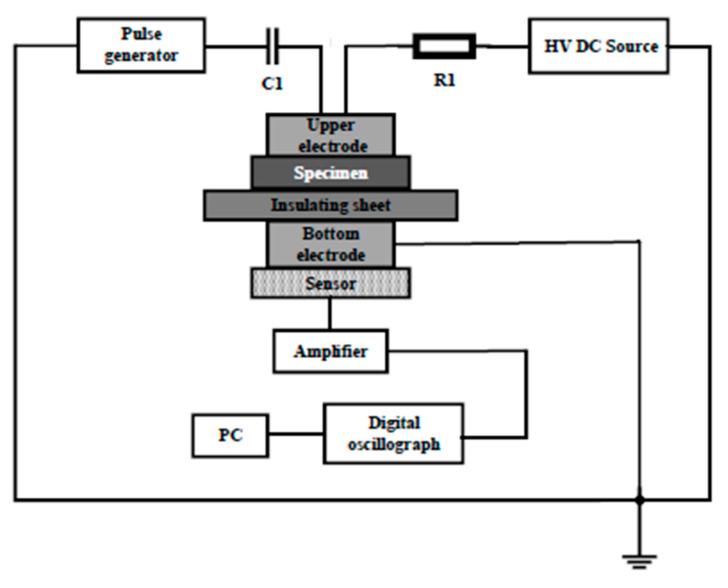
Schematic diagram of the pulse electro-acoustic (PEA) method.

**Figure 4 polymers-12-00809-f004:**
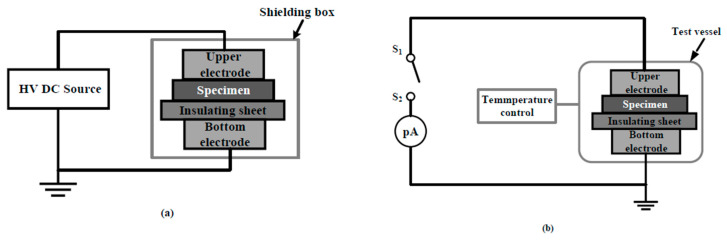
Schematic diagram of the thermal-stimulation depolarization currents (TSDC) experiment setup. (**a**) Polarization setup; (**b**) schematic of Novocontrol system.

**Figure 5 polymers-12-00809-f005:**
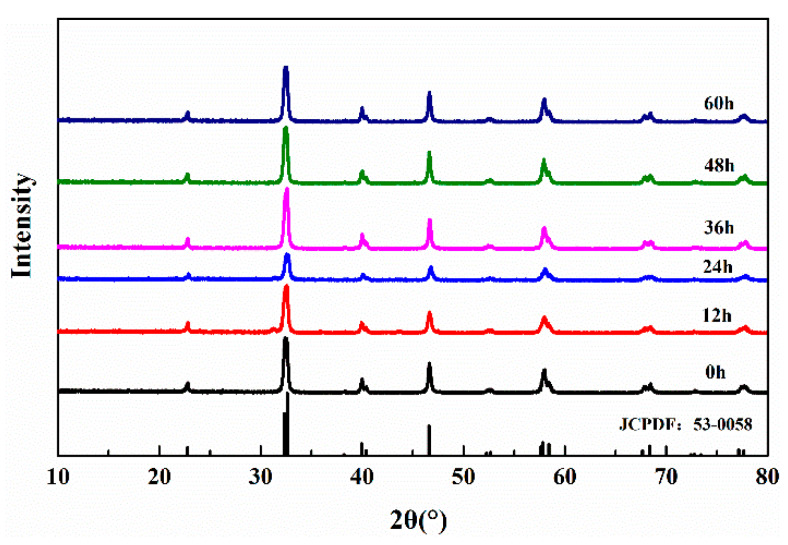
XRD patterns of La_0.8_Sr_0.2_MnO_3_ powders under different ball milling durations.

**Figure 6 polymers-12-00809-f006:**
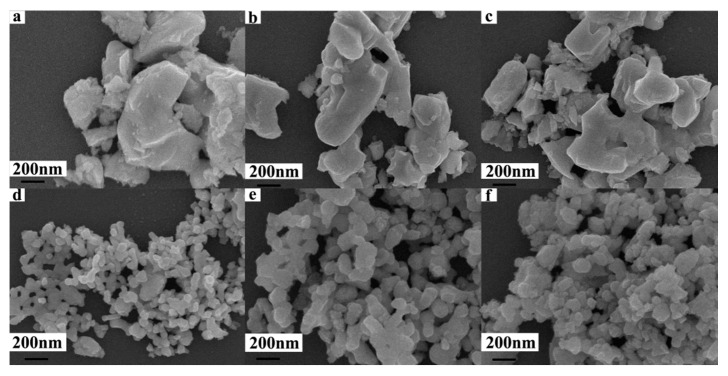
FE-SEM images of La_0.8_Sr_0.2_MnO_3_ powers. (**a**) LSM (0 h), (**b**) BM-LSM (12 h), (**c**) BM-LSM (24 h), (**d**) BM-LSM (36 h), (**e**) BM-LSM (48 h), (**f**) BM-LSM (60 h). BM = ball-milled.

**Figure 7 polymers-12-00809-f007:**
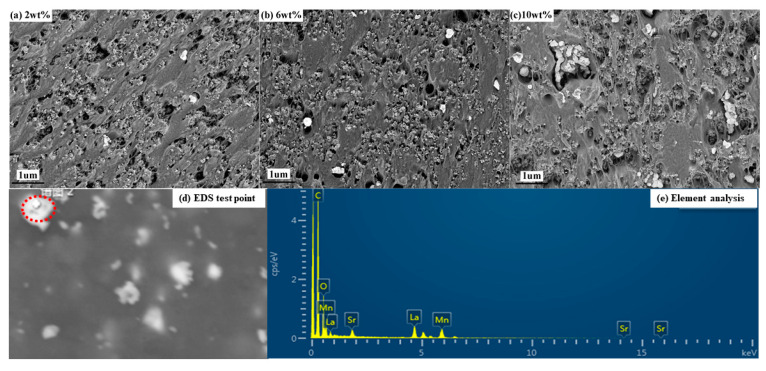
The SEM images and element analysis of La_0.8_S_r0.2_MnO_3_/semi-conductive composites. (**a**) LSM content 2 wt. %; (**b**) LSM content 6 wt. %; (**c**) LSM content 10 wt. %; (**d**) sampling point of EDS test (**e**) element analysis.

**Figure 8 polymers-12-00809-f008:**
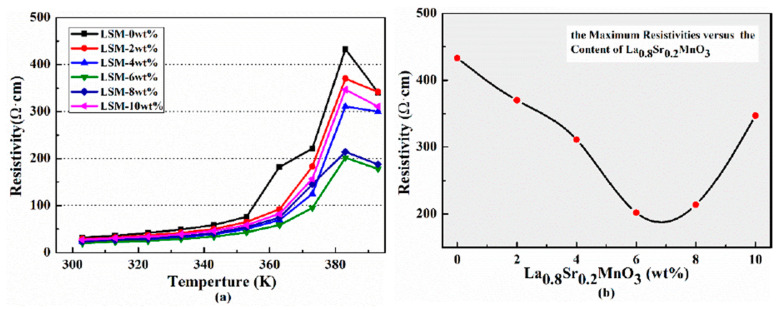
Resistivity-temperature behaviors of La_0.8_S_r0.2_MnO_3_/semi-conductive composites. (**a**) Resistivity of La_0.8_Sr_0.2_MnO_3_/semi-conductive composites versus temperature; (**b**) maximum resistivity versus La_0.6_Sr_0.4_MnO_3_ content.

**Figure 9 polymers-12-00809-f009:**
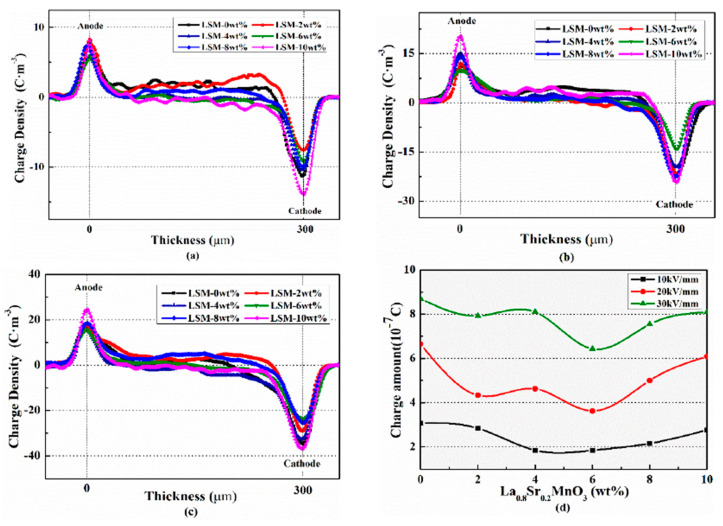
Comparison of space charge distributions at 1800 s in the insulation samples under semi-conductive composites with different La_0.8_S_r0.2_MnO_3_ contents. (**a**) 10 kV/mm; (**b**) 20 kV/mm; (**c**) 30 kV/mm; (**d**) space charge amount.

**Figure 10 polymers-12-00809-f010:**
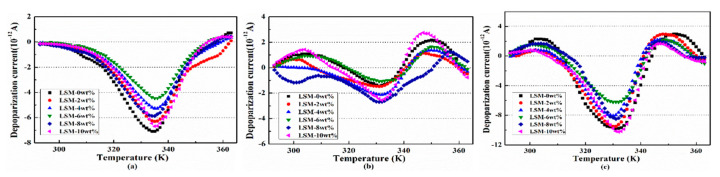
Depolarization current in insulation under semi-conductive composites with different La_0.8_S_r0.2_MnO_3_ contents. (**a**) 10 kV/mm; (**b**) 20 kV/mm; (**c**) 30 kV/mm.

**Figure 11 polymers-12-00809-f011:**
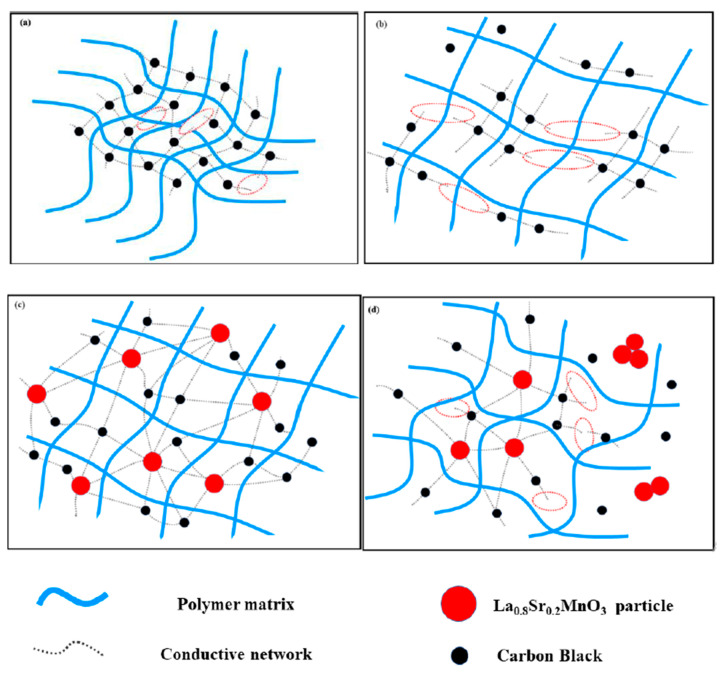
The schematic diagram of the forming of conductive network in different semi-conductive composites at different temperatures. (**a**) Semi-conductive composites without La_0.8_Sr_0.2_MnO_3_ at room temperature; (**b**) semi-conductive composites without La_0.8_Sr_0.2_MnO_3_ Near the melting point; (**c**) semi-conductive composites with 6 wt.% La_0.8_Sr_0.2_MnO_3_ near the melting point; (**d**) semi-conductive composites with 10 wt. % La_0.8_Sr_0.2_MnO_3_ near the melting point.

**Figure 12 polymers-12-00809-f012:**
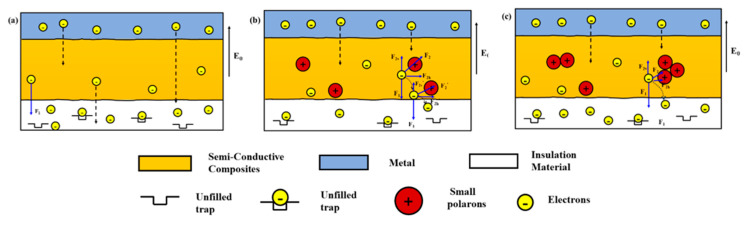
Schematic diagram of the electrons moving path in the semi-conductive layer under the combined action of electric field and the small polarons. (**a**) Without La_0.8_Sr_0.2_MnO_3_; (**b**) La_0.8_Sr_0.2_MnO_3_ content 6 wt. %; (**c**) La_0.8_Sr_0.2_MnO_3_ content exceed 6 wt. %.

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
