# Peer review of "Synthesis of a Novel Semi-Conductive Composites Doping with La_0.8_Sr_0.2_MnO_3_ for Excellent Electric Performance for HVDC Cable"

_polymers, 2020, doi:10.3390/polym12040809_

Round 1

Reviewer 1 Report

The manuscript "Synthesis of A Novel Semi -Conductive Composites Doping with La0.8Sr0.2MnO3 for Excellent Electric Performance for HVDC Cable" describes a method using  semi conductive composites doped with different amount of LaSrMnO3 to reduce space charge accumulation in HVDC caples. So far there flaws in the manuscript which need be addressed

If using abreviation such as PTC in abstract please define where it first appears.

The manuscript has a lots of figures which not needed as example Figure 5 of ball milling the Figure 6 where the images nearly similiar. Some should be shown in supplementary.

There no real discussion provided nor any references given to other works done before. This is not acceptable for a scientific work. The authors have to provide a proper discussion.

At figure 7 they shown sem cross section a to c where no real differences can be seen. As well there is another sem images not even mention in figure title. The EDX spectroscopy should identify all signals and between 4 to 6 keV between La and Mn peak there 2 other not specify. Please verify.

At Figure 8

the authors notify a negative temperature coefficients

"Furthermore, the NTC effect occurs at 393 K much higher than the working temperature of the HVDC cable. Thus, the phenomenon can be ignored"

There only one measurement made or one signal taken where in all samples a negative temperature coefficient shown. It can not be understood why such phenomen can be ignored and no reference state such statement. It also would be interesting what happen if temperature further increased.

The authors also claim that a 10 degrees increase in HVDC caples is a great effect. Since they shown K instead of celcius a 10K increase surely not great improvement.

Figure 8b shows the resistivity against content of LaSr MnO3 content. There no explanation given why the resistivity decrease with 8wt% to 10 wt%. Please give a reasonable explanation with underlying literature and discussion.

Author Response

Point 1: If using abreviation such as PTC in abstract please define where it first appears. Response 1:I have define all the abreciations in where they first appear, HVDC appears at line 2 , LSM appears at line 14, PTC and PEA appears at line 15, TSDC appears at line 16, LDPE appears at line 36, XLPE appears at line 37, CB and EVA appears at line 42, XRD appears at line 115, FE-SEM appears at line 116, EDS appears at line 185, NTC appears at line 243. Point 2: The manuscript has a lots of figures which not needed as example Figure 5 of ball milling the Figure 6 where the images nearly similiar. Some should be shown in supplementary. Response 2: Figure 5 is the XRD patterns of LSM powder with different ball milling time.It can be confirmed from the figure that the crystal form of LSM is not damaged by ball milling. Figure 6 is the SEM images of La0.8Sr0.2MnO3 powder with different ball milling time. It can be observed from the morphology of LSM particles, we can choose the shortest ball-milling time to make the size of LSM particles uniform. It can be concluded from figure 5 and 6 that the sintered La0.8Sr0.2MnO3 sample can turned into a nano-powder with size of 100-200nm by ball milling at 300 rpm for 48 hours.(It can be seen in the manuscript line 208-209) Point 3:There no real discussion provided nor any references given to other works done before. This is not acceptable for a scientific work. The authors have to provide a proper discussion. Response 3:I have added discussion section3.3 and sectionv4. Point 4:At figure 7 they shown sem cross section a to c where no real differences can be seen. As well there is another sem images not even mention in figure title. The EDX spectroscopy should identify all signals and between 4 to 6 keV between La and Mn peak there 2 other not specify. Please verify. Response 4:I have reworked Figure 7 and added a title to Figure 7 d.This is a common method for rubber or plastic composites to observe the cross-section micromorphology. Figure 7 d and e is an element dispersive spectrum(EDS). In figure e the unmarked spectral signals between 4-6Kev is the Lβ and Lγ spectral line of element La, the marked signal is Lα of La. According to standard spectrum library, Lα locates at 4.65KeV, Lβ and Lγ locates at 5.04KeV and 5.19KeV. The intensity ratio between them is100:70:10. Usually only the strongest Lα is marked. Point 5:At Figure 8 the authors notify a negative temperature coefficients "Furthermore, the NTC effect occurs at 393 K much higher than the working temperature of the HVDC cable. Thus, the phenomenon can be ignored" .There only one measurement made or one signal taken where in all samples a negative temperature coefficient shown. It can not be understood why such phenomen can be ignored and no reference state such statement. It also would be interesting what happen if temperature further increased.The authors also claim that a 10 degrees increase in HVDC caples is a great effect. Since they shown K instead of celcius a 10K increase surely not great improvement. Response 5:Because the melt range of LDPE is 110-115℃ and the melt point of EVA 7470 is 76℃. Furthermore, during the preparation of the seim-conductive composite, we also learned that the polymers have melted at 120℃. Moreover, considering its insulation is polyethylene, the operating temperature of HVDC cable can not exceed 363K[5,6]. Since the NTC effect occurs at 393 K which is much higher than the operational temperature of the HVDC cable. Thus, the phenomenon can be ignored for HVDC cable. I have interpret the PTC effect and NTC effect in section4 line 316-336. Point 6:Figure 8b shows the resistivity against content of La0.8Sr0.2MnO3 content. There no explanation given why the resistivity decrease with 8wt% to 10 wt%. Please give a reasonable explanation with underlying literature and discussion. Response 6:When LSM content is 6wt% the maximum resistivity of the specimen is 202Ω﹒㎝, when LSM content is 4wt% the maximum resistivity is 311Ω﹒㎝, when LSM content is 8wt% the maximum resistivityis 214Ω﹒㎝ and when LSM content is 10wt% the maximum resistivity is 347Ω﹒㎝.The resistivity does not decrease with 8wt% to 10 wt%.

Reviewer 2 Report

The authors have carried out an original study. They have evaluated the influence of semi-conductive polymers modified with La0.8Sr0.2MnO3 on PTC effect suppress and charge injection characteristics in the insulation sample were studied.

However, some details could be explain clearly, for example, when the content of La0.8Sr0.2MnO3  is 6wt.%, PTC effect and charge injection are both be suppressed most effectively, whereas this content is about 10wt.%, PTC effect suppression is not as obviously as that of 6wt.% and charge suppression effect is weakened.

Your reason is due to the distribution of the inorganic conductive particles at the interface between the crystalline phase and the amorphous phase. That is Ok but where you have justified and/or found these results?

It is difficult to follow your interpretations for example, you have discussed (section 4) some of the results that you have obtained in the section 3 during the results and discussion? In the section 3.3 Resistivity of La0.8Sr0.2MnO3/Semi-conductive Composites???

I suggest you to simplify mainly for the first readers your interpretations (section 4) and please simplify also your conclusion that you have summarized your results

Author Response

Point 1: However, some details could be explain clearly, for example, when the content of La0.8Sr0.2MnO3  is 6wt.%, PTC effect and charge injection are both be suppressed most effectively, whereas this content is about 10wt.%, PTC effect suppression is not as obviously as that of 6wt.% and charge suppression effect is weakened.

Your reason is due to the distribution of the inorganic conductive particles at the interface between the crystalline phase and the amorphous phase. That is OK but where you have justified and/or found these results?

Response 1: I think it can be justify from figure 7a-c. I have changed more sharper images about the  microstructure of the corss-section of the semi-conductive sepcimens with La0.8Sr0.2MnO3 contents of 2wt.%, 6 wt.% and 10wt.% respectively. It can be seen from the images that the ionic particles scattered at the interface between the crystalline phase and the amorphous phase. When La0.8Sr0.2MnO3 contents of 2wt.%, 6 wt.%, particles size is evenly and small, however, when La0.8Sr0.2MnO3 content is 10wt.%,  the particle size become more bigger, and it can be seen clearly that  the grains has agglomerated.

Point 2: It is difficult to follow your interpretations, for example, you have discussed (section 4) some of the results that you have obtained in the section 3 during the results and discussion? In the section 3.3 Resistivity of La0.8Sr0.2MnO3/Semi-conductive Composites???

I suggest you to simplify mainly for the first readers your interpretations (section 4) and please simplify also your conclusion that you have summarized your results

Response 2: I have asjusted the content of section 3and 4. Some discussion has been adjusted and added in section 4, Line 311-335 is for the PTC effect ,line 360-3423 is for the effect of the Coulomb force from LSM on moving charges.

Round 2

Reviewer 1 Report

All questions sufficient answered